# Strong and Efficient Baselines for
# Open Domain Conversational Question Answering

**Andrei C. Coman**[*]
Idiap Research Institute, EPFL
andrei.coman@idiap.ch

**Gianni Barlacchi**
Amazon Alexa AI
gbarlac@amazon.com

**Adrià de Gispert**
Amazon Alexa AI
agispert@amazon.com

## Abstract

Unlike the Open Domain Question Answering (ODQA) setting, the conversational (OD-ConvQA) domain has received limited attention when it comes to reevaluating baselines for both efficiency and effectiveness. In this paper, we study the State-of-the-Art (SotA) Dense Passage Retrieval (DPR) retriever and Fusion-in-Decoder (FiD) reader pipeline, and show that it significantly underperforms when applied to ODConvQA tasks due to various limitations. We then propose and evaluate strong yet simple and efficient baselines, by introducing a fast reranking component between the retriever and the reader, and by performing targeted finetuning steps. Experiments on two ODConvQA tasks, namely TopiOCQA and OR-QuAC, show that our method improves the SotA results, while reducing reader's latency by 60%. Finally, we provide new and valuable insights into the development of challenging baselines that serve as a reference for future, more intricate approaches, including those that leverage Large Language Models (LLMs).

## 1 Introduction

In an automated information-seeking conversation scenario between two parties, the human questioner asks a series of questions and expects to receive a relevant response from the answering system (Oddy, 1977). Current State-of-the-Art (SotA) shapes the answerer via two neural models, the Dense Passage Retrieval (DPR) (Karpukhin et al., 2020) and the Fusion-in-Decoder (FiD) (Izacard and Grave, 2021b), which act as retriever and reader, respectively. Their success stems from the ability to overcome certain limitations of their sparse and extractive counterparts, such as not relying on lexical retrieval heuristics or extracting spans as a response (Chen et al., 2017; Yang et al., 2019; Lee et al., 2019; McCallum et al., 2019; Guu et al., 2020; Lewis et al., 2020; Shen et al., 2023).

Among the most promising approaches are those concerning the improvement of training strategies (Guu et al., 2020; Balachandran et al., 2021; Qu et al., 2021), use of rerankers (Hu et al., 2019; Mao et al., 2021; Barlacchi et al., 2022; Iyer et al., 2021; Glass et al., 2022), question rewriting (Vakulenko et al., 2021; Del Tredici et al., 2021), reader to retriever knowledge distillation (Izacard and Grave, 2021a), memory-efficient pipeline (Izacard et al., 2020; Del Tredici et al., 2022), and leveraging structured information (Min et al., 2019; Yu et al., 2022).

Unlike the Open Domain Question Answering (ODQA) setting, a reassessment of the baselines in terms of both efficiency and effectiveness appears to be under-explored in the conversational (OD-ConvQA) domain. In this paper, we focus on the typical DPR retriever and FiD reader (DPR+FiD) pipeline, and show its limitations when applied to the ODConvQA setting. Despite its popularity, we find that this baseline significantly underperforms when finetuned on downstream tasks. We show that simple improvements in the training, architecture, and inference setups of the DPR+FiD pipeline, provide a strong and efficient baseline that exceeds the performance of SotA models on two common ODConvQA datasets: TopiOCQA (Adlakha et al., 2022) and ORConvQA (OR-QuAC) (Qu et al., 2020).

We point out several limitations of the pipeline, such as: 1) *reader's susceptibility to noisy input*, 2) *retriever's reduced coverage*, 3) *retriever's lack of cross semantic encoding between the conversation and the retrieved passages*, and 4) *reader's latency is heavily impacted by the number of input passages*. To mitigate these, we propose and evaluate a simple and effective approach by including a fast reranking component between the retriever and the reader, and by performing targeted finetuning steps. The proposed Retriever-Reranker-Reader finetuning (R3FINE) strategy leads to baseline models with a better latency/performance trade-off.

---

[*]Work was done during an internship at Amazon Science.

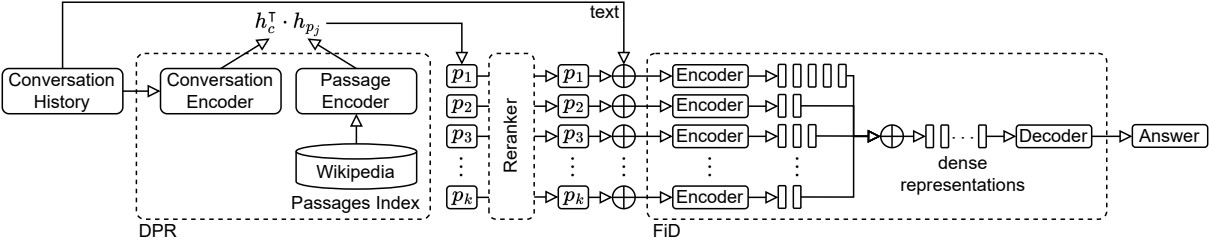

Figure 1: The Retriever-Reranker-Reader (R3) pipeline

These baselines, which are simple and easy to replicate, serve as a reference point for comparing new and more complex models, and determining their effectiveness. Our contributions are the following:

- We identify and address several limitations of the typical pipeline used in ODConvQA.

- We propose the R3FINE strategy, which improves SotA results on two common datasets and reduces pipeline's latency by 60%.

- We provide new and valuable insights for creating simple and efficient baselines, which serve as a reference point for future comparison of new more complex approaches.

## 2   End-to-End Baselines for ODConvQA

This section provides a brief introduction to the pipeline on which this work focuses. Figure 1 shows the typical pipeline used within the OD-ConvQA setting, featuring an additional reranker component. A conversation history is input to the DPR retriever. This module exploits a dual-encoder based on the BERT (Devlin et al., 2019) model. First, it encodes the conversation history via the $ConversationEncoder$ component, which takes as input the text of the conversation history $c_1, c_2, \ldots, c_i$, and then it outputs a dense representation $h_c$. Next, this representation is used to perform a dense search to retrieve the most relevant passages, i.e., text blocks that serve as basic retrieval units, from an external knowledge source (e.g., Wikipedia). The latter contains dense representations of the passages that have been encoded via the $PassageEncoder$ component, which takes as input a $j$-th passage with a given text length $N$, i.e., $p_{j_1}, p_{j_2}, \ldots, p_{j_N}$, and outputs a dense representation $h_{p_j}$. The dense search is performed via the Maximum Inner-Product Search (MIPS) function which outputs the value corresponding to $h_c^\intercal \cdot h_{p_j}$.

Once the top-$k$ relevant passages have been retrieved, their text is appended to the conversa-

tion history and subsequently passed to the FiD reader, which is based on the T5 (Raffel et al., 2020) model. The newly created textual sequences of length $S$ are then encoded in parallel via the $Encoder$ component that outputs a dense representation $\bar{h} = \{h_1, \ldots, h_S\}$. As a final step, the dense representations of the entire list of $k$ input passages are concatenated to form a single $\bar{h}_1 \oplus \bar{h}_2, \ldots, \oplus \bar{h}_k$ sequence that forms the input to the $Decoder$ component responsible for generating the answer $a$.

## 3   Strong Baseline Models

This work focuses on two main datasets. TOPIOCQA (Adlakha et al., 2022) is a large-scale open-domain information-seeking conversational dataset that contains a challenging phenomenon in the form of topic switching. OR-QuAC (Qu et al., 2020) leverages CANARD's (Elgohary et al., 2019) context-independent question rewrites of the QuAC (Choi et al., 2018) dataset, and adapts it to the open-domain setting. Further details regarding the datasets are provided in Appendix A.

We outline a number of limitations of the typical DPR+FiD pipeline, along with suggestions on how to mitigate them. While some of those interconnect at different levels the various efforts made in the ODQA domain (Balachandran et al., 2021; Yu et al., 2022), our goal is to offer a perspective on the ODConvQA setting.

### 3.1   Current Limitations and Bottlenecks

**Reader's susceptibility to noisy input.**   Previous findings have shown that the FiD reader performance significantly improves when increasing the number of retrieved passages (Izacard and Grave, 2021b). While confirming this finding, in Table 1 we also present a different perspective to it. We show that when the same reader model is provided with the relevant (i.e., gold) passage in input, the performance decreases as the number of retrieved passages increases. This suggests that there is a balance in presenting input to the reader: if the gold

| | TOPIOCQA | | | |
|---|---|---|---|---|
| | w/o gold | | w/ gold | |
| top-$k$ | EM | F1 | EM | F1 |
| 1 | 19.3 | 37.6 | 38.3 | 65.5 |
| 10 | 29.8 | 52.4 | 35.8 | 61.5 |
| 50 | 33.0 | 55.1 | 35.9 | 59.5 |

Table 1: FiD reader performance (Exact Match and F1 scores) on the TOPIOCQA dev split, with/without the gold passage (w/o gold) in the top-$k$ limit.

| | TOPIOCQA | | OR-QuAC | |
|---|---|---|---|---|
| top-$k$ | w/o SR | w/ SR | w/o SR | w/ SR |
| 1 | 24.66 | 42.64 | 29.27 | 56.64 |
| 10 | 62.45 | 75.62 | 57.14 | 72.27 |
| 50 | 77.41 | 84.69 | 65.22 | 73.53 |
| 1000 | 91.49 | 91.49 | 74.55 | 74.55 |

Table 2: Dev split retrieval coverage before/after the introduction of the $SemanticReranker$ (w/o SR) when a larger number of passages is considered (50 vs 1000).

passage is present, i.e., the retriever could retrieve it, a small relevant list is best, but otherwise a larger list is better.

**Retriever's reduced coverage.** Current solutions impose a hard top-$k$ limit on the number of passages returned by the DPR retriever and assume that the relevant ones are present within this limit. Table 1 shows that coverage is key during the retrieval phase for the reader to perform well. To improve it, we suggest introducing a simple and efficient Transformer-based (Vaswani et al., 2017) reranker component after the retriever. This component, shown in Figure 1 and described in the next paragraph, is designed to reconsider a larger pool of passages returned by the DPR and to provide the FiD with a reduced and improved list of passages. Since this module operates at the semantic level, we refer to it as the $SemanticReranker$. Table 2 shows the potential coverage margins and the retrieval results obtained after the introduction of such a module, when a larger number of passages (50 vs 1000) is considered.

**Retriever's lack of cross semantic encoding between the conversation and the retrieved passages.** The DPR retriever performs independent encoding of the passages via the $PassageEncoder$ function. This means that it is not able to exploit the semantic relationship among them. This can be mitigated via the introduction of the previously mentioned $SemanticReranker$ component. This new module is based on the *TransformerEncoder* and applies the following function:

$$\tilde{h}_c, \tilde{h}_{p_1}, \ldots, \tilde{h}_{p_k} = Reranker(h_c, h_{p_1}, \ldots, h_{p_k})$$

where each element of the input attends to both the conversation dense representation $h_c$ and passages dense representations $h_{p_i}$. Reranking is performed over the new output sequence $\tilde{h}_c, \tilde{h}_{p_1}, \ldots, \tilde{h}_{p_k}$ via the previously mentioned MIPS function.

**Reader's latency is heavily impacted by the number of input passages.** Figure 2 shows that the latency of the reader can be significantly reduced by decreasing the number of input passages. However, a trivial limitation to top-$k$ considerably degrades the performance of the module, thus leading to an inevitable trade-off. The task of the $SemanticReranker$ involves pushing relevant passages into the top-$k$ list, and allowing for a low $k$ value to be set.

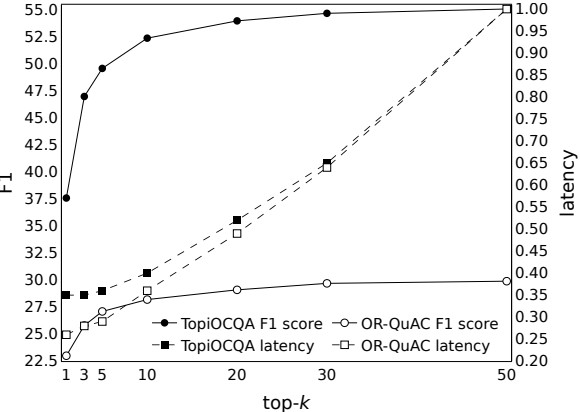

Figure 2: FiD reader performance (F1 score and latency) on the TOPIOCQA dev split and OR-QuAC test split, with varying top-$k$ input passages. Latency is relative to the top-50 (top-$k$ vs top-50).

### 3.2 A Strong and Efficient Baseline

Based on the findings above, we introduce the Retriever-Reranker-Reader finetuning (R3FINE) strategy, which can be used to design strong and efficient baselines for ODConvQA. First, we increase the number of passages returned by the DPR from the initial 50 to 1000. Then, we add the $SemanticReranker$ component, which corresponds to a single $TransformerEncoder$ layer. We train/finetune the $SemanticReranker$ along with the $ConversationEncoder$ while keeping the $PassageEncoder$ frozen. Guided by the intu-

| | TOPIOCQA | | | | | | OR-QuAC | | | | | |
|---|---|---|---|---|---|---|---|---|---|---|---|---|
| | w/o SR | | w/ SR | | w/ SR + FT | | w/o SR | | w/ SR | | w/ SR + FT | |
| top-$k$ | EM | F1 | EM | F1 | EM | F1 | EM | F1 | EM | F1 | EM | F1 |
| 1 | 19.3 | 37.6 | 28.1 | 50.4 | 30.7 | 52.4 | 14.7 | 23.0 | 13.9 | 25.7 | 16.6 | 28.9 |
| 10 | 29.8 | 52.4 | 33.2 | 57.3 | **35.8** | **59.0** | 19.0 | 28.2 | 19.4 | 30.0 | 21.6 | **32.9** |
| 50 | 33.0 | 55.1 | 33.9 | 56.2 | 35.2 | 56.1 | 22.0 | 29.9 | 22.1 | 30.2 | **23.5** | 32.2 |

Table 3: FiD reader performance (Exact Match and F1 scores) on the TOPIOCQA dev split and OR-QuAC test split before/after the introduction of the $SemanticReranker$ (w/o SR), together with the results obtained after a further reader finetuning step with top-10 output by the SR (w/ SR + FT). Underlined values indicate the results obtained by the DPR+FiD pipeline. **Bold** values indicate the results obtained after the introduction of the SR together with targeted finetuning steps.

| TOPIOCQA | | |
|---|---|---|
| Model | EM | F1 |
| BM25 + DPR Reader | 13.6 | 25.0 |
| BM25 + FiD | 24.1 | 37.2 |
| DPR Retriever + DPR Reader | 21.0 | 43.4 |
| DPR Retriever + FiD | 33.0 | 55.3 |
| (Ours) DPR Retriever + FiD | 33.0 | 55.1 |
| + R3FINE (top-10) | **35.8** | **59.0** |

Table 4: TOPIOCQA dev split baselines performance (Exact Match and F1 scores) comparison between sparse/dense retrievers (BM25/DPR Retriever) and extractive/generative readers (DPR Reader/FiD).

| OR-QuAC | | |
|---|---|---|
| Model | EM | F1 |
| DrQA (Chen et al., 2017) | - | 6.3 |
| BERTserini (Yang et al., 2019) | - | 26.0 |
| ORConvQA (Qu et al., 2020) | - | 29.4 |
| (Ours) DPR Retriever + FiD | **22.0** | 29.9 |
| + R3FINE (top-10) | 21.6 | **32.9** |

Table 5: OR-QuAC test split baselines performance (Exact Match and F1 scores) comparison.

ition that less but more relevant passages are beneficial to FiD as reported in Table 1, we finally perform an additional finetuning step by leveraging the new top-10 list of passages returned by the $SemanticReranker$.

## 4 Experiments and Results

This section shows the impact that the introduction of the $SemanticReranker$ module has on the pipeline, as well as the finetuning steps we followed to make the pipeline more efficient without compromising its performance.

**Experimental Setup.** As the starting point of our experiments, we used the DPR and FiD models provided with the TOPIOCQA dataset. Currently, only the *train* and *dev* splits are made available for this dataset. We followed the same experimental setup and exploited TOPIOCQA's DPR module for both datasets. Unlike TOPIOCQA, OR-QuAC is of extractive type, and for this reason we trained the FiD module from scratch by following the same training configuration of TOPIOCQA.

**End-to-End Results.** Table 4 and Table 5 compare our R3FINE strategy with previous baselines.

R3FINE achieves an F1 score of 59 points on TOPIOCQA, and 32.9 on OR-QuAC, which are 3.9 and 3 points higher than the best models proposed in the original papers. It is worth noting that these large improvements are achieved with simple adjustments in the training, architecture, and inference setups of the well-established DPR+FiD baseline, and not via the introduction of new heavier and complex models.

To further support our R3FINE strategy, in Table 3 we present an ablation study which quantifies its impact on the DPR+FiD pipeline. We note that introducing the $SemanticReranker$ (w/ SR) always outperforms the DPR+FiD baseline (w/o SR), and at the same time it allows for a 5-fold input size reduction (top-10) while obtaining on-par or better results. In addition, a further finetuning step of the FiD (w/ SR + FT) outperforms the results obtained by the $SemanticReranker$ (w/ SR) by 1.7 and 2.9 F1 points on TOPIOCQA and OR-QuAC, respectively. Further experiments and ablation studies are provided in Appendix A.

Finally, in Figure 2 it can also be observed that using top-10 instead of top-50 can reduce FiD's latency by 60% on average across the two datasets. We conducted a latency measurement to evaluate the impact of the $SemanticReranker$

and its associated parameters, with detailed information available in Appendix A. Given that the $SemanticReranker$ consists of a single $TransformerEncoder$ layer, its parameters are negligible when compared to both the DPR and FiD. Moreover, the $SemanticReranker$ accounts only for 0.34% of the overall latency of the FiD reader, adding an additional 2.4ms per example on top of the 710ms taken by FiD. It is important to note that this impact is only considered in relation to FiD, as the retrieval phase remains constant regardless of the inclusion of the $SemanticReranker$.

## 5 Conclusions

In this paper, we identified several limitations of the typical Depnse Passage Retrieval (DPR) retriever and Fusion-in-Decoder (FiD) reader pipeline when applied in an ODConvQA setting. We proposed and evaluated an improved approach by including a fast reranking component between these two modules and by performing targeted finetuning steps. The proposed R3FINE strategy lead to a better latency/performance trade-off. The new baseline has proven to be both strong and efficient when compared to previous baselines, thus making it suitable for future comparisons of new approaches.

## Limitations

The study presented in this work aimed to identify and address various limitations of the commonly used ODConvQA pipeline. While our approach may not be technically groundbreaking, the work's novelty lies in the presented findings to design strong and efficient baselines for ODConvQA. It should be noted that further research is needed to compare the performance of the proposed R3FINE strategy with other rerankers on non-conversational QA datasets, which would provide valuable insights into how effective the R3FINE approach is in different contexts.

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

# A  Appendix

This section provides additional information in support of the work done within the paper.

## A.1  OR-QuAC conversion to TOPIOCQA's format and models

To make the OR-QuAC dataset compatible with TOPIOCQA's models, we applied the following steps:

- all answers of type CANNOTANSWER and NOTRECOVERED have been mapped to UNANSWERABLE

- for the DPR's $ConversationEncoder$ component, we followed the TOPIOCQA's ALL-HISTORY conversation representation

- for the DPR's $PassageEncoder$ component, each passage title has been reduced from "passage_page_title [SEP] passage_page_subtitle" to "passage_page_title". This is due to the fact that, compared to TOPIOCQA, OR-QuAC does not provide the information about the section where a particular passage is located within the page.

- for the FiD component, each passage information has been reduced from "title: sub-title: context:" to "title: context:". This is done for the same reason mentioned in the previous point.

Each passage text in OR-QuAC's Wikipedia knowledge source has been mapped to its corresponding embedding via the TOPIOCQA's $PassageEncoder$ component. We then performed the same retrieval step as the one done for TOPIOCQA. We exploited TOPIOCQA's DPR module for both datasets as the retrieval phase is very similar between the two. However, given that, unlike TOPIOCQA, OR-QuAC is of extractive type, we had to train the FiD module from scratch. We followed the same training configuration as the one used for TOPIOCQA.

## A.2  Reranker training and ablation study

We tried different configurations of the $SemanticReranker$ to find the most efficient and effective one. In addition to the decision of whether to finetune the DPR's $ConversationEncoder$ together with the $SemanticReranker$, we also tried varying the number of layers $L$ of the $SemanticReranker$ from 1 to 4 and changing its input, by choosing a combination from:

- $h_{c_1}$: use of conversation's history dense representation

- $h_{c_1}, \ldots, h_{c_i}$: use of conversation's history tokens dense representation

- $p_1, \ldots, p_k$: use of passages dense representation

Table 6 shows the average top-$k$ results obtained on the TOPIOCQA and OR-QuAC dev split, where $k$ varies between 1, 3, 5, 10, 15, 20, 30, 50, 100, 250, 500, 750, and 1000. For TOPIOCQA, we report the presence of the gold passage within the top-$k$ limit. For OR-QuAC we report the presence of the gold answer within the top-$k$ limit.

Given that the MIPS function cannot be applied when $p_1, \ldots, p_k$ representations are used alone, i.e, without the conversation history, we applied a linear projection on top of the $SemanticReranker$ to obtain a score for each passage in input.

As far as training the $SemanticReranker$ is concerned, we trained it for 10 epochs when finetuned together with the $ConversationEncoder$. We instead trained it for 20 epochs when the $ConversationEncoder$ was kept frozen and when $p_1, \ldots, p_k$ representations were used alone. We leveraged the same objective function used for training the initial DPR. We used early stopping to chose the best performing model on the dev set. We also used a linear learning rate decay throughout the training process, and AdamW with a learning rate of 5e-5 and weight decay of 1e-2.

Among the different combinations shown in Table 6, we considered the first entry as the best choice, i.e., the model with $L = 1$, $h_{c_1}$, $p_1, \ldots, p_k$, and $CrossEncoder$ finetuning.

## A.3  Retriever results

Table 7 shows the train split retrieval coverage before/after the introduction of the $SemanticReranker$ (w/o SR) when a larger number of passages is considered (50 vs 1000).

| $L$ | $h_{c_1}$ | $h_{c_1},\ldots,h_{c_i}$ | $p_1,\ldots,p_k$ | $CrossEncoder$ | TOPIOCQA | OR-QuAC |
|---|---|---|---|---|---|---|
| 1 | ✓ | | ✓ | ✓ | 78.85 | 71.51 |
| 1 | | ✓ | ✓ | ✓ | 78.88 | 71.48 |
| 4 | | ✓ | ✓ | ✓ | 78.84 | 71.32 |
| 1 | ✓ | | ✓ | | 77.17 | 69.83 |
| 1 | | ✓ | ✓ | | 77.35 | 70.18 |
| 4 | | ✓ | ✓ | | 77.30 | 70.45 |
| 1 | | | ✓ | | 67.07 | 64.18 |

Table 6: Average top-$k$ results obtained on the TOPIOCQA and OR-QuAC dev split, with different configurations.

For both datasets, we report the presence of the gold passage within the top-$k$ limit.

| | TOPIOCQA | | OR-QuAC | |
|---|---|---|---|---|
| top-$k$ | w/o SR | w/ SR | w/o SR | w/ SR |
| 1 | 31.54 | 98.25 | 31.69 | 77.31 |
| 5 | 66.99 | 99.71 | 59.08 | 89.10 |
| 10 | 78.29 | 99.72 | 66.31 | 89.36 |
| 20 | 86.78 | 99.72 | 71.77 | 89.40 |
| 50 | 93.55 | 99.72 | 77.35 | 89.40 |
| 500 | 99.36 | 99.72 | 87.15 | 89.41 |
| 1000 | 99.72 | 99.72 | 89.41 | 89.41 |

Table 7: Train split retrieval coverage before/after the introduction of the $SemanticReranker$ (w/o SR) when a larger number of passages is considered (50 vs 1000). For both datasets, we report the presence of the gold passage within the top-$k$ limit.

Table 8 shows the dev split retrieval coverage before/after the introduction of the $SemanticReranker$ (w/o SR) when a larger number of passages is considered (50 vs 1000). For TOPIOCQA, we report the presence of the gold passage within the top-$k$ limit. For OR-QuAC we report the presence of the gold answer within the top-$k$ limit. Table 9 shows the OR-QuAC test split retrieval coverage before/after the introduction of the $SemanticReranker$ (w/o SR) when a larger number of passages is considered (50 vs 1000). We report the presence of the gold answer within the top-$k$ limit.

| | TOPIOCQA | | OR-QuAC | |
|---|---|---|---|---|
| top-$k$ | w/o SR | w/ SR | w/o SR | w/ SR |
| 1 | 24.66 | 42.64 | 29.27 | 56.64 |
| 5 | 51.87 | 68.62 | 51.63 | 70.82 |
| 10 | 62.45 | 75.62 | 57.14 | 72.27 |
| 20 | 70.21 | 80.75 | 61.37 | 72.80 |
| 50 | 77.41 | 84.69 | 65.22 | 73.53 |
| 500 | 89.58 | 90.73 | 72.97 | 74.34 |
| 1000 | 91.49 | 91.49 | 74.55 | 74.55 |

Table 8: Dev split retrieval coverage before/after the introduction of the $SemanticReranker$ (w/o SR) when a larger number of passages is considered (50 vs 1000). For TOPIOCQA, we report the presence of the gold passage within the top-$k$ limit. For OR-QuAC we report the presence of the gold answer within the top-$k$ limit.

| | OR-QuAC | |
|---|---|---|
| top-$k$ | w/o SR | w/ SR |
| 1 | 26.82 | 48.75 |
| 5 | 46.29 | 64.71 |
| 10 | 51.50 | 66.47 |
| 20 | 55.63 | 67.31 |
| 50 | 59.85 | 68.10 |
| 500 | 67.87 | 69.48 |
| 1000 | 69.86 | 69.86 |

Table 9: OR-QuAC test split retrieval coverage before/after the introduction of the $SemanticReranker$ (w/o SR) when a larger number of passages is considered (50 vs 1000). We report the presence of the gold answer within the top-$k$ limit.

### A.4 Reader results

Table 10, Table 11, and Table 12 show the impact the introduction of the $SemanticReranker$ has on the FiD reader. The input to the FiD reader are either passages returned by the initial DPR retriever (w/o SR) or passages returned by the $SemanticReranker$ (w/ SR).

### A.5 Reader is susceptible to noisy input

Table 13 shows the FiD reader performance on the TOPIOCQA dev split, with/without the gold passage (w/o gold) in the top-$k$ limit. This analysis is limited to the TOPIOCQA dataset as it is the only one to provide information about the gold passage for the dev split.

| TOPIOCQA | | | | |
|---|---|---|---|---|
| | w/o SR | | w/ SR | |
| top-$k$ | EM | F1 | EM | F1 |
| 1 | 19.3 | 37.6 | 28.1 | 50.4 |
| 5 | 27.0 | 49.6 | 32.2 | 56.5 |
| 10 | 29.8 | 52.4 | 33.2 | 57.3 |
| 20 | 31.3 | 54.0 | 33.4 | 56.5 |
| 50 | 33.0 | 55.1 | 33.9 | 56.2 |

Table 10: FiD reader performance (Exact Match and F1 scores) on the TOPIOCQA dev split before/after the introduction of the $SemanticReranker$ (w/o SR).

| OR-QuAC | | | | |
|---|---|---|---|---|
| | w/o SR | | w/ SR | |
| top-$k$ | EM | F1 | EM | F1 |
| 1 | 13.2 | 22.2 | 12.2 | 25.9 |
| 5 | 16.4 | 26.3 | 16.4 | 28.7 |
| 10 | 18.0 | 27.6 | 18.0 | 29.4 |
| 20 | 19.2 | 28.2 | 18.9 | 29.2 |
| 50 | 19.6 | 27.7 | 19.3 | 27.7 |

Table 11: FiD reader performance (Exact Match and F1 scores) on the OR-QuAC dev split before/after the introduction of the $SemanticReranker$ (w/o SR).

| OR-QuAC | | | | |
|---|---|---|---|---|
| | w/o SR | | w/ SR | |
| top-$k$ | EM | F1 | EM | F1 |
| 1 | 14.7 | 23.0 | 13.9 | 25.7 |
| 5 | 17.8 | 27.1 | 17.9 | 29.5 |
| 10 | 19.0 | 28.2 | 19.4 | 30.0 |
| 20 | 20.3 | 29.1 | 20.8 | 30.5 |
| 50 | 22.0 | 29.9 | 22.1 | 30.2 |

Table 12: FiD reader performance (Exact Match and F1 scores) on the OR-QuAC test split before/after the introduction of the $SemanticReranker$ (w/o SR).

| TOPIOCQA | | | | |
|---|---|---|---|---|
| | w/o gold | | w/ gold | |
| top-$k$ | EM | F1 | EM | F1 |
| 1 | 19.3 | 37.6 | 38.3 | 65.5 |
| 5 | 27.0 | 49.6 | 36.3 | 62.5 |
| 10 | 29.8 | 52.4 | 35.8 | 61.5 |
| 20 | 31.3 | 54.0 | 36.2 | 60.8 |
| 50 | 33.0 | 55.1 | 35.9 | 59.5 |

Table 13: FiD reader performance (Exact Match and F1 scores) on the TOPIOCQA dev split, with/without the gold passage (w/o gold) in the top-$k$ limit.

## A.6 Further reader study

To better understand the impact that the introduction of the $SemanticReranker$ has on the FiD reader, Table 14, Table 15, and Table 16 show the results obtained after taking a non-finetuned FiD and training it on the top-10 passages returned by the initial DPR retriever and on the top-10 passages returned by the $SemanticReranker$. On both datasets, we followed the same training configuration as the one used for TOPIOCQA.

| TOPIOCQA | | | | |
|---|---|---|---|---|
| | w/o SR | | w/ SR | |
| top-$k$ | EM | F1 | EM | F1 |
| 1 | 19.2 | 37.8 | 29.5 | 50.9 |
| 5 | 27.8 | 49.9 | 34.0 | 56.9 |
| 10 | 30.4 | 52.0 | 34.3 | 57.0 |

Table 14: FiD reader performance (Exact Match and F1 scores) on the TOPIOCQA dev split after taking a non-finetuned FiD and training it on the top-10 passages returned by the initial DPR retriever (w/o SR) and on the top-10 passages returned by the $SemanticReranker$ (w/ SR).

| OR-QuAC | | | | |
|---|---|---|---|---|
| | w/o SR | | w/ SR | |
| top-$k$ | EM | F1 | EM | F1 |
| 1 | 14.2 | 23.1 | 13.9 | 27.6 |
| 5 | 18.0 | 27.7 | 18.1 | 30.8 |
| 10 | 19.2 | 28.9 | 19.3 | 31.2 |

Table 15: FiD reader performance (Exact Match and F1 scores) on the OR-QuAC dev split after taking a non-finetuned FiD and training it on the top-10 returned by the initial DPR retriever (w/o SR) and on the top-10 returned by the $SemanticReranker$ (w/ SR).

| OR-QuAC | | | | |
|---|---|---|---|---|
| | w/o SR | | w/ SR | |
| top-$k$ | EM | F1 | EM | F1 |
| 1 | 15.5 | 23.9 | 15.8 | 28.3 |
| 5 | 19.6 | 29.1 | 19.4 | 31.7 |
| 10 | 20.9 | 29.7 | 20.6 | 31.9 |

Table 16: FiD reader performance (Exact Match and F1 scores) on the OR-QuAC test split after taking a non-finetuned FiD and training it on the top-10 passages returned by the initial DPR retriever (w/o SR) and on the top-10 passages returned by the $SemanticReranker$ (w/ SR).

Table 17, Table 18, and Table 19 show instead

the results obtained after taking an already fine-tuned FiD reader and further finetuning it on the top-10 passages returned by the initial DPR retriever and on the top-10 passages returned by the $SemanticReranker$. On both datasets, the amount of finetuning steps is equal to the one used for training the already finetuned FiD reader.

| | w/o SR | | w/ SR | |
|---|---|---|---|---|
| **top-$k$** | **EM** | **F1** | **EM** | **F1** |
| 1 | 21.5 | 39.3 | 30.7 | 52.4 |
| 5 | 30.4 | 52.3 | 36.1 | 59.4 |
| 10 | 32.8 | 54.6 | 35.8 | 59.0 |

**TOPIOCQA**

Table 17: FiD reader performance (Exact Match and F1 scores) on the TOPIOCQA dev split after taking an already finetuned FiD and further finetuning it on the top-10 returned by the initial DPR retriever (w/o SR) and on the top-10 returned by the $SemanticReranker$ (w/ SR).

| | w/o SR | | w/ SR | |
|---|---|---|---|---|
| **top-$k$** | **EM** | **F1** | **EM** | **F1** |
| 1 | 15.2 | 23.8 | 15.4 | 29.0 |
| 5 | 18.9 | 28.4 | 18.4 | 31.2 |
| 10 | 19.9 | 29.4 | 20.1 | 32.0 |

**OR-QuAC**

Table 18: FiD reader performance (Exact Match and F1 scores) on the OR-QuAC dev split after taking an already finetuned FiD and further finetuning it on the top-10 passages returned by the initial DPR retriever (w/o SR) and on the top-10 passages returned by the $SemanticReranker$ (w/ SR).

| | w/o SR | | w/ SR | |
|---|---|---|---|---|
| **top-$k$** | **EM** | **F1** | **EM** | **F1** |
| 1 | 16.7 | 24.9 | 16.6 | 28.9 |
| 5 | 20.0 | 29.2 | 20.1 | 32.2 |
| 10 | 21.4 | 30.0 | 21.6 | 32.9 |

**OR-QuAC**

Table 19: FiD reader performance (Exact Match and F1 scores) on the OR-QuAC test split after taking an already finetuned FiD and further finetuning it on the top-10 passages returned by the initial DPR retriever (w/o SR) and on the top-10 passages returned by the $SemanticReranker$ (w/ SR).

### A.7 Latency measurement

Latency measurement (see Figure 2) has been performed on the same NVIDIA V100 16GB GPU, by following the FiD's `test_reader.py` script provided with the TOPIOCQA dataset. We set the `per_gpu_batch_size` paramenter to 4 in all runs and chose the value of the `n_context` parameter from 1, 3, 5, 10, 20, 30, and 50, based on the number of input passages. For each value, we report the latency relative to the maximum `n_context` parameter value, i.e., 50. We used CUDA events synchronization markers to measure the elapsed time for the preprocessing and evaluation of TOPIOCQA's dev split and OR-QuAC's test split.