# OpenReview forum: "Strong and Efficient Baselines for Open Domain Conversational Question Answering"
_EMNLP/2023/Conference — EMNLP 2023 Findings_

### Official Review · Reviewer_tKUu · 2023-08-04

**Soundness:** 4

**Excitement:**

2: Mediocre: This paper makes marginal contributions (vs non-contemporaneous work), so I would rather not see it in the conference.

**Missing References:**

1. Glass, M.R., Rossiello, G., Chowdhury, M.F., Naik, A.R., Cai, P., & Gliozzo, A. (2022). Re2G: Retrieve, Rerank, Generate. ArXiv, abs/2207.06300.

**Paper Topic And Main Contributions:**

The paper focuses on open-domain conversational question answering. It proposes adding a fast reranker between the retriever and the reader to balance speed and performance. It claims new SOTA results on two benchmarks and a 60% reduction in latency.

**Questions For The Authors:**

Q1A. Line 15 and line 187 says that the latency is about the decoder, while line 80 says that the latency is about the pipeline, which in my understanding, is retriever+reader. Which one is correct?

Q1B. If the 60% reduction of latency is about the reader, then what is the overall reduction of latency and how does the reranker slow down the pipeline? If is about the pipeline, then what is cost of each component?

Q2. What is the size of T5 used in your experiments? And what value is the length S? Are there any computational issues when experiments with k=50, especially during training, because k=50 means a very long input?

**Reasons To Accept:**

1. This paper provides four reasonable limitations on the retriever-reader framework.

2. There are sufficient experiments to support all their claims and the effectiveness of their model.

3. There are results on many alternative designs.


**Reasons To Reject:**

1. I cannot see any significant difference between the R3FINE (proposed in this paper) and Re2G[1]. Re2G is evaluated on Wizard of Wikipedia, another open-domain conversational QA benchmark (not used in this paper), and shows the effectiveness of the retriever-reranker-reader framework on ODConvQA. Re2G also discussed the efficiency benefits brought by the reranker.

2. Actually, many ODConvQA models use rerankers. Except for Re2G, the baseline model in this paper, ORConvQA, is also equipped with a reranker, although the reranker in ORConvQA does not aim to improve efficiency. However, this paper does not provide any discussion on this.

3. This paper distinguishes itself from open-domain QA (non-conversational). But I cannot find a strong reason why rerankers in ODQA differ from rerankers in ODConvQA.

[1] Glass, M.R., Rossiello, G., Chowdhury, M.F., Naik, A.R., Cai, P., & Gliozzo, A. (2022). Re2G: Retrieve, Rerank, Generate. ArXiv, abs/2207.06300.

**Reproducibility:**

4: Could mostly reproduce the results, but there may be some variation because of sample variance or minor variations in their interpretation of the protocol or method.

**Reviewer Confidence:**

2: Willing to defend my evaluation, but it is fairly likely that I missed some details, didn't understand some central points, or can't be sure about the novelty of the work.

**Typos Grammar Style And Presentation Improvements:**

L002: DPR and FiD are not abbreviations of their preceding words.

---

> ### Author Rebuttal · Authors · 2023-08-28
>
> We genuinely thank the reviewer for their constructive input, which aligns with our efforts to enhance the robustness of our work. Their acknowledgment of the four limitations we pinpointed for the retriever-reader framework, the substantial experimental validation supporting our assertions, and our thorough exploration of alternative designs underscore the rigor and comprehensiveness of our research.
>
> __Reviewer__
>
> *"Line 15 and line 187 says that the latency is about the decoder, while line 80 says that the latency is about the pipeline, which in my understanding, is retriever+reader. Which one is correct?"*
>
> *"If the 60% reduction of latency is about the reader, then what is the overall reduction of latency and how does the reranker slow down the pipeline? If is about the pipeline, then what is cost of each component?"*
>
> __Authors__
>
> Given that the SemanticReranker consists of a single TransformerEncoder layer, its parameters are negligible when compared to both the DPR and FiD. Moreover, the SemanticReranker accounts for 0.34% of the overall latency of the FiD reader, adding an additional 2.4ms per example on top of the 710ms taken by FiD. It is important to note that this impact is only considered in relation to FiD, as the retrieval phase remains constant regardless of the inclusion of the SemanticReranker. We will make sure to provide these additional details in the camera-ready.
>
> __Reviewer__
>
> *"What is the size of T5 used in your experiments? And what value is the length S? Are there any computational issues when experiments with k=50, especially during training, because k=50 means a very long input?"*
>
> __Authors__
>
> To ensure complete compatibility with the conventional DPR+FiD pipeline, we follow the same experimental setup of the TopiOCQA dataset, where the FiD reader consists of a T5-base model. It is important to note that the FiD reader does not encode information from multiple passages in a cross-semantic way during the encoding phase. Instead, for efficiency reasons, each passage is encoded independently and in parallel with the conversation history. The cross-semantic connections between passages are only established during the decoding phase. A common FiD reader uses 50 passages, but our approach allows the reranker to work with 1000 passages and simultaneously condition on both passages and conversation. This leads to increased coverage (Table 2) and improved overall performance (Table 4 and 5). Given the limitations of the reader component, the use of a reranker appears to be necessary. Additionally, the proposed reranker helps to significantly reduce the latency of the reader without compromising its performance (Table 3 and Figure 2).
>
> __Reviewer__
>
> *“I cannot see any significant difference between the R3FINE (proposed in this paper) and Re2G [1]. […]”*
>
> __Authors__
>
> We will diligently assess the missing reference highlighted by the reviewer, ensuring that proper recognition is accorded where it is due. We believe that the novelty of our work lies in the presented findings on how to design strong and efficient baselines for ConvQA. We argue that it is not limited to the simple addition of a reranker. In fact, we demonstrate that the typical pipeline used in the ODConvQA setting exhibits several limitations and that current baselines are not being utilized to their full potential, which can lead to unreliable conclusions when developing new models. To mitigate the problem, we propose simple improvements in training, architecture, and inference that lead to stronger and more efficient baselines that surpass current state-of-the-art models (Table 1 and 3 and Figure 2).
>
> __Reviewer__
>
> *”This paper distinguishes itself from open-domain QA (non-conversational). But I cannot find a strong reason why rerankers in ODQA differ from rerankers in ODConvQA.”*
>
> __Authors__
>
> In this work, we decided to focus mainly on the Open Domain Conversational Question Answering (ODConvQA) domain as we identified a lack of exploration into the efficiency and effectiveness of the typical DPR+FiD pipeline within this setting. Our goal was to keep the paper concise and focused on a specific setting where the limitations play an important role. This implied the exclusion of experiments on ODQA and their relegation as future work.

---

### Official Review · Reviewer_3qL2 · 2023-08-04

**Soundness:** 3

**Excitement:**

3: Ambivalent: It has merits (e.g., it reports state-of-the-art results, the idea is nice), but there are key weaknesses (e.g., it describes incremental work), and it can significantly benefit from another round of revision. However, I won't object to accepting it if my co-reviewers champion it.

**Paper Topic And Main Contributions:**

This study sheds light on the limitations of the dense passage retrieval + fusion-in-decoder architecture in open-domain conversational QA and proposes a new approach to overcome these limitations. The problems include that the reader should have fewer inputs as long as the input contains gold passages for both performance and latency reasons, and that the output of the retriever does not semantically take the conversation history into account sufficiently. Their proposed method adds a lightweight model that reranks the passages output by the retriever, taking into account the conversation. The authors experimentally showed that the proposed method improves performance and reduces latency on two datasets.


**Questions For The Authors:**

1. Lines 144-148: To clarify, are the "w/" and "w/o gold" results based on different evaluation data for the "same" model (rather than two models trained on different data)?
2. Lines 251-254: Figure 2 does not discuss the latency that SemanticReranker brings. Please discuss the impact that the introduction of it has had.


**Reasons To Accept:**

1. The proposed method not only experimentally shows the problems of the DPR+FiD architecture but also improves the performance by adding a reranker. Furthermore, the effectiveness of the added components was demonstrated through an ablation study. This simple idea may be of interest to other researchers working on open-domain conversational QA.


**Reasons To Reject:**

1. Although the proposed method significantly improves performance, this paper does not explain how the hyper-parameters are set. If they were tuned with evaluation data in a non-systematic manner, the advantages of the proposed method might be limited in other settings. However, the basic idea of the proposed method seems promising to me, and the lack of comprehensive experiments should not be the main reason for rejecting a short paper.


**Reproducibility:**

4: Could mostly reproduce the results, but there may be some variation because of sample variance or minor variations in their interpretation of the protocol or method.

**Reviewer Confidence:**

3: Pretty sure, but there's a chance I missed something. Although I have a good feel for this area in general, I did not carefully check the paper's details, e.g., the math, experimental design, or novelty.

**Typos Grammar Style And Presentation Improvements:**

Presentation

- Lines 087-089: Figure 1 illustrates a proposed method, not a "typical pipeline", doesn't it?
- Section 2: The symbols used here are complicated, and this paragraph should be reorganised in places. If I understand correctly, (a) the conversation history (line 095) may be written as c instead of c_1, ..., c_i, (b) a j-th passage (line 103) should be introduced with the symbol p_j, and the reference to its text length, N, and p_j_N (which makes confusion with p_1, ..., p_N) may be omitted in the text, and (c) a dense representation \overline{h} can be explained without {h_1, ..., h_S}.
- Figure 1: The notations here should follow the symbols in Section 2. In other words, c, S, h_1, ..., h_k, and a should be mentioned in the figure. Also, the Reranker should take as input the outputs of the conversation encoder in addition to p_1, ..., p_n. The output of the Reranker should be p_k instead of p_n.
- Lines 149-151: (i.e., if the retriever could return the ideal answers)
- Line 167: The term "SemanticReranker" should also be introduced here, as referenced in Table 2.
- Line 183: Is the \tilde{h}_c different from h_c? If so, it is not obvious to me how different.

Typos

- Line 546: the that the -> the
- Line 583: 30 might be included here, I think.

---

> ### Author Rebuttal · Authors · 2023-08-28
>
> We would like to thank the the reviewer for their thorough evaluation and the constructive remarks provided regarding the R3FINE method. We appreciate their acknowledgment of the method's effectiveness and efficiency, along with their recognition of the experimental validity and the potential impact of our contributions. Their meticulous analysis and helpful suggestions for improving the clarity of our presentation are greatly appreciated, and we are dedicated to promptly addressing these recommendations.
>
> __Reviewer__
>
> *”Lines 251-254: Figure 2 does not discuss the latency that SemanticReranker brings. Please discuss the impact that the introduction of it has had.”*
>
> __Authors__
>
> Given that the SemanticReranker consists of a single TransformerEncoder layer, its parameters are negligible when compared to both the DPR and FiD. Moreover, the SemanticReranker accounts for 0.34% of the overall latency of the FiD reader, adding an additional 2.4ms per example on top of the 710ms taken by FiD. It is important to note that this impact is only considered in relation to FiD, as the retrieval phase remains constant regardless of the inclusion of the SemanticReranker. We will make sure to provide these additional details in the camera-ready.
>
> __Reviewer__
>
> *"Lines 144-148: To clarify, are the "w/" and "w/o gold" results based on different evaluation data for the "same" model (rather than two models trained on different data)?"*
>
> __Authors__
>
> In Table 1, and its more comprehensive counterpart Table 13 (Appendix A.5), the goal is to show that the “same” FiD reader is susceptible to noisy input. The terms “w/” and “w/o gold” denote results that are indeed based on different evaluation data for the “same” model. To be more precise, the variation between these two settings is solely dependent on whether the gold passage is included within the top-k limit or not.
>
> __Reviewer__
>
> *"Line 183: Is the \tilde{h}_c different from h_c? If so, it is not obvious to me how different."*
>
> __Authors__
>
> Given that the SemanticReranker leverages the TransformerEncoder layer, its output vectors represent a mixture of its input vectors. The inclusion of the notation serves the purpose of accentuating this distinction.

---

### Official Review · Reviewer_UNK4 · 2023-08-05

**Soundness:** 3

**Excitement:**

3: Ambivalent: It has merits (e.g., it reports state-of-the-art results, the idea is nice), but there are key weaknesses (e.g., it describes incremental work), and it can significantly benefit from another round of revision. However, I won't object to accepting it if my co-reviewers champion it.

**Paper Topic And Main Contributions:**

This paper points out four limitations of the existing pipeline and proposes Retriever-Reranker-Reader fine-tuning strategy. First, they increase the number of passages returned by the DPR from 50 to 1000. Then they add a Semantic Reranker component, which is fine-tuned with the Conversation Encoder. Their proposed method improves SOTA results on two common ODConVQA datasets: TopiOCQA and ORConvQA and reduces the pipeline's latency by 60%.

**Questions For The Authors:**

A. In Table 4, I was wondering about the difference between the method DPR Retriever + FiD and (Ours) DPR Retriever + FiD. They have the same EM score while the latter has a lower F1 score.

**Reasons To Accept:**

The paper is well-written and clear. The proposed method is simple and effective. The experiments conducted on two datasets support the main claims.

**Reasons To Reject:**

Inconsistent results. On OR-QuAC dataset in Table 5, the EM score decreases from 22.0 to 21.6 while F1 increases from 29.9 to 32.9 after adding the SR. This is different from the performance on TOPIOCQA in Table 4, where both EM and F1 increase.

**Reproducibility:**

3: Could reproduce the results with some difficulty. The settings of parameters are underspecified or subjectively determined; the training/evaluation data are not widely available.

**Reviewer Confidence:**

3: Pretty sure, but there's a chance I missed something. Although I have a good feel for this area in general, I did not carefully check the paper's details, e.g., the math, experimental design, or novelty.

---

> ### Author Rebuttal · Authors · 2023-08-28
>
> We appreciate the reviewer's insightful evaluation and supportive feedback on our approach, highlighting the paper's clarity, the method's simplicity and effectiveness, and the robust experimental evidence.
>
> __Reviewer__
>
> *"Inconsistent results. On OR-QuAC dataset in Table 5, the EM score decreases from 22.0 to 21.6 while F1 increases from 29.9 to 32.9 after adding the SR. This is different from the performance on TOPIOCQA in Table 4, where both EM and F1 increase."*
>
> __Authors__
>
> Table 5 shows a slight decline in the EM performance of the "+ R3FINE (top-10)" model, which is attributed to its use of only 10 passages, in contrast to the "(Ours) DPR Retriever + FiD" baseline, which uses 50 passages. Nevertheless, Table 3's "w/ SR + FT" column indicates that the R3FINE model surpasses the baselines in both metrics when given access to the same number of passages (i.e., 50). For the OR-QuAC dataset, we opted for consistency in reporting results on both EM and F1 metrics, aligning with our approach for TopiOCQA. This decision has been made to establish a standardized reporting framework for future research, thereby enhancing the comparability of different models.
>
> __Reviewer__
>
> *"In Table 4, I was wondering about the difference between the method DPR Retriever + FiD and (Ours) DPR Retriever + FiD. They have the same EM score while the latter has a lower F1 score."*
>
> __Authors__
>
> The tag "(Ours)" indicates a model that leverages the pre-trained models from prior research, yet incorporates a distinct codebase and pipeline implementation compared to the original approach. This decision was primarily driven by the necessity to seamlessly integrate the SemanticReranker into the existing framework. The marginal reduction in the F1 score may be attributed to minor variances in the retrieved passages and their ranking. These nuances can marginally influence the answer generation process.

---

### Meta-Review · Senior_Area_Chairs · 2023-10-05

**Recommendation:** 3

**Metareview:**

The paper points at weaknesses in the retriever-reader framework for open domain conversational QA, and suggests a retriever-reranker-reader approach that shows benefits on two benchmark datasets.
While the approach is not very novel (there were previous works that suggested reranking in similar settings, e.g. as pointed by reviewer 3), the experiments are sound and demonstrate appealing results as agreed by all reviewers.
Given the above, the paper may be accepted to findings.

---

### Decision · Program_Chairs · 2023-10-07

**Decision:**

Accept-Findings

**Comment:**

The paper points at weaknesses in the retriever-reader framework for open domain conversational QA, and suggests a retriever-reranker-reader approach that shows benefits on two benchmark datasets.
While the approach is not very novel (there were previous works that suggested reranking in similar settings, e.g. as pointed by reviewer 3), the experiments are sound and demonstrate appealing results as agreed by all reviewers.
Given the above, the paper may be accepted to findings.